# DTO-KD: Dynamic Trade-off Optimization for Effective Knowledge Distillation

**Zeeshan Hayder**[1,2]**, Ali Cheraghian**[1,2]**, Lars Petersson**[1]**, Mehrtash Harandi**[3]**, Richard Hartley**[2]
[1]Data61 / CSIRO, Australia
[2]Australian National University (ANU), Australia
[3]Monash University, Australia

## Abstract

Knowledge Distillation (KD) is a widely adopted framework for compressing large models into compact student models by transferring knowledge from a high-capacity teacher. Despite its success, KD presents two persistent challenges: (1) the trade-off between optimizing for the primary task loss and mimicking the teacher's outputs, and (2) the gradient disparity arising from architectural and representational mismatches between teacher and student models. In this work, we propose Dynamic Trade-off Optimization for Knowledge Distillation (DTO-KD), a principled multi-objective optimization formulation of KD that dynamically balances task and distillation losses at the gradient level. Specifically, DTO-KD resolves two critical issues in gradient-based KD optimization: (i) gradient conflict, where task and distillation gradients are directionally misaligned, and (ii) gradient dominance, where one objective suppresses learning progress on the other. Our method adapts per-iteration trade-offs by leveraging gradient projection techniques to ensure balanced and constructive updates. We evaluate DTO-KD on large-scale benchmarks including ImageNet-1K for classification and COCO for object detection. Across both tasks, DTO-KD outperforms prior KD methods, yielding state-of-the-art accuracy and improved convergence behavior. Furthermore, student trained with DTO-KD exceed the performance of their non-distilled counterparts, demonstrating the efficacy of our multi-objective formulation.

## 1 Introduction

Deep neural networks have demonstrated impressive performance across a wide range of computer vision tasks; however, their practical deployment is often hindered by substantial computational and memory demands, particularly on resource-limited platforms such as edge devices and mobile hardware. This has motivated increasing interest in model compression techniques that reduce network complexity without sacrificing accuracy. Among these, knowledge distillation (KD) (Yim, 2017; Gao et al., 2018; Qiu et al., 2023; Zhou et al., 2020) has emerged as an effective strategy, in which a compact student model is trained under the guidance of a large, pre-trained teacher model. By transferring informative representations or outputs from the teacher, the student can achieve comparable performance with significantly fewer parameters, making it more amenable to deployment in constrained environments. A standard KD framework typically combines a task-specific objective (e.g., classification or detection loss) with an auxiliary distillation loss that facilitates knowledge transfer from the teacher to the student.

Early studies on knowledge distillation (KD) (Hinton et al., 2015; Zhang et al., 2018) primarily relied on training the student model using the teacher's output predictions as supervisory signals. While effective to some extent, this strategy is inherently limited, as the teacher's final outputs provide a highly compressed representation of its knowledge, and relying solely on logits constrains the richness of transferable information. To overcome these shortcomings, subsequent KD methods (Romero et al., 2015; Chen et al., 2020; Heo et al., 2019a) moved toward leveraging intermediate feature representations from the teacher, allowing for more expressive and flexible knowledge transfer. Such feature-based distillation methods often depend on heuristic design choices and introduce additional hyperparameters that require careful, task-dependent tuning. Nevertheless, despite these improvements, recent feature-level KD approaches (Chen et al., 2022; 2021; Roy Miles & Deng,

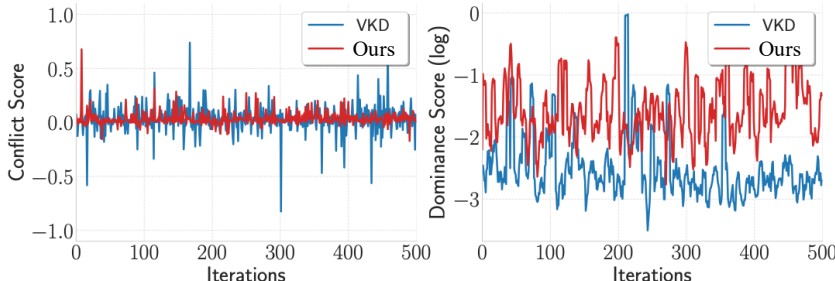

Figure 1: **Gradient Dynamics analysis**, comparing the conflict and dominance behavior of the distillation and task gradients. **Left**) Conflict score is computed as $\langle g_{\text{dist}}, g_{\text{task}} \rangle$, where more negative values indicate stronger disagreement. **Right**) Dominance score is calculated as $\frac{|g_{\text{dist}}|}{|g_{\text{task}}|}$ and shown in log-scale, with lower values indicating stronger dominance. DTO-KD achieves lower gradient conflict and more balanced gradient dominance compared to the baseline.

2024) continue to face challenges in effectively distilling knowledge from highly expressive teacher networks into compact student models, largely due to mismatches between the objectives imposed by ground-truth supervision and those induced by the distillation process.

Recent KD methods (Wang et al., 2024; Chen et al., 2022) propose heuristic mechanisms for balancing teacher mimicking, yet optimization inconsistency persists as a critical bottleneck to efficient knowledge transfer. The primary issue limiting the performance of these approaches is two-fold. First, **Gradient Conflicts (GrC)** arise when the gradients of the task-specific objective and the distillation process are misaligned. Second, **Gradient Dominance (GrD)** occurs when the gradient magnitude of one objective (e.g., either distillation or task-specific) dominates the learning process, causing an imbalance. Figure 1 illustrates these issues by plotting gradient conflict and dominance for our method and that of Roy Miles & Deng (2024) over 500 iterations on the detection task.

To address all of these issues, we propose a novel distillation optimization strategy. Specifically, we frame the problem as a dynamic trade-off optimization, which not only efficiently resolves gradient conflicts during training but also ensures a Pareto optimal (Lin, 1976) solution. This results in a training strategy that eliminates the need for manually tuning hyperparameters to balance the contributions of each loss function. Instead, it dynamically learns the contribution of each loss function, adapting between task-specific and distillation-specific objectives throughout the training.

To be more specific, in this paper we propose a closed-form method for determining how to weight the distillation and task-specific losses during training. Unlike the prior work of (Liu et al., 2023), our approach provides an explicit solution that can be computed efficiently at each step. In teacher–student architectures, where the distillation and task losses evolve rapidly, existing task-weighting methods (Hu et al., 2024; Zheng & Yang, 2024) can struggle to adapt, causing weights to oscillate or lag behind the changing dynamics. In contrast, our closed-form solution produces an update direction that is jointly aligned with both objectives, ensuring that neither the distillation nor the task loss dominates or interferes with the other. As a result, our method naturally mitigates gradient conflict and yields a more stable and effective multi-objective learning process.

In this paper, we introduce DTO-KD (Dynamic Trade-off Optimization for Knowledge Distillation), a novel multi-objective learning framework that formulates knowledge distillation as a gradient-level optimization problem. DTO-KD improves the efficiency and effectiveness of knowledge transfer by dynamically modulating the contribution of task-specific and distillation-specific objectives during training, removing the need for manual loss weighting or extensive hyperparameter tuning. DTO-KD is trained end to end and demonstrates faster convergence, requiring fewer epochs to reach or exceed the performance of state-of-the-art distillation methods. In summary, the contributions of this paper are as follows:

- We propose DTO-KD, a dynamic trade-off optimization framework that balances task and distillation losses at the gradient level. This principled approach eliminates the need for fixed loss weighting, enabling adaptive trade-offs during training.
- DTO-KD resolves gradient conflict (GrC) and dominance (GrD) via per-iteration gradient balancing approach, leading to aligned, balanced updates and improved convergence.

- We conduct extensive experiments on both classification and detection benchmarks, achieving state-of-the-art performance. Ablation studies confirm the robustness of DTO-KD across diverse distillation setups.

## 2 RELATED WORK

This section explores KD techniques, focusing on the use of logits, CNN features, and transformer features (or tokens). Additionally, it examines multi-objective approaches relevant to the DTO-KD.

**Aligning Predictive Distributions via Logit-Level Distillation:** Classical Knowledge Distillation (KD) primarily relies on the teacher's output logits. Ensemble-based collaborative learning (Zhang et al., 2018), multi-stage distillation with teacher assistants (Mirzadeh et al., 2020), and decoupled distillation across teacher branches (Zhao et al., 2022a) all fall within this paradigm. Most methods use forward KL-divergence to align teacher-student distributions, though this often yields overly smoothed predictions. Reverse KL-divergence (Wang et al., 2025a) emphasizes the teacher's high-confidence modes, while $\alpha$–$\beta$ divergence (Wang et al., 2025b) generalizes both by interpolating between their behaviors. However, logit-based approaches transfer only final-layer information, omitting rich intermediate representations and teacher inductive biases. Consequently, students struggle to reconcile teacher outputs with task-specific objectives, often leading to limited generalization.

**Transferring Intermediate Representations via Feature-Level Distillation:** Feature-based Knowledge Distillation (KD) leverages intermediate activations to convey structural knowledge unavailable at the logit level (Yang et al., 2021; Xu et al., 2020). Early work such as FitNets (Romero et al., 2015) introduced stage-wise alignment, enabling the student to mimic deeper teacher features. Subsequent methods refine how features are selected and aligned. Margin ReLU filtering (Heo et al., 2019a) suppresses redundant activations and improves feature matching, while analysis of connection pathways (Chen et al., 2021) highlights the importance of optimal teacher–student layer mappings. Diffusion-based distillation (Huang et al., 2024) reduces noise in student features before transferring knowledge, and norm/direction-aware losses (Wang et al, 2024) further enhance feature alignment. Despite progress, feature-level KD still struggles to encode long-range dependencies and global contextual knowledge that is essential in modern architectures and increasingly captured by transformer-based models.

**Leveraging Transformer Semantics via Token-Level Distillation:** Transformer-based Knowledge Distillation (KD) focuses on transferring information encoded in self-attention and token interactions. DeiT (Touvron et al., 2022) first established efficient token distillation for convolution-free vision transformers. For detection tasks, token-matching methods (Song et al., 2021; 2022) require the student to replicate teacher tokens, though naive token alignment is often insufficient. More advanced methods incorporate multiple teachers (Ren et al., 2022), manifold-based token alignment (Hao et al., 2022), and generalized f-divergence formulations (Wen et al., 2023) that flexibly weight the teacher's dominant predictions across tokens. However, transferring dark knowledge, such as subtle interactions and contextual semantics encoded in token relations, remains challenging. Approaches such as non-target logit normalization (Yang et al., 2023) or two-stage early-layer distillation pipelines (Chen et al., 2022) offer partial solutions but remain heuristic and fragmented. In contrast, our work proposes a unified end-to-end formulation that dynamically balances objectives through trade-off optimization.

**Aligning Conflicting Objectives via Multi-objective Optimization:** Multi-objective optimization (MOO) enables simultaneous optimization of conflicting objectives by seeking Pareto-optimal trade-offs. A simple approach re-weights loss functions based on manually designed criteria (Chen et al., 2018; Kendall et al., 2018), but these methods are often heuristic, ignore dynamic gradient interactions, and lack strong theoretical foundations. Gradient manipulation methods (Sener & Koltun, 2018; Yu et al., 2020; Liu et al., 2021b;a; 2023) instead combine gradients from different tasks at each step. For example, Sener & Koltun (2018) uses an upper bound for efficiency, Yu et al. (2020) projects gradients to avoid conflicts, Liu et al. (2021b;a) provide a closed-form solution minimizing average loss, and Liu et al. (2023) introduces a fast dynamic weighting method. Although MOO is explored in multi-task learning, In this paper, we proposed DTO-KD which uniquely applies it to knowledge distillation, formulating it as a dynamic trade-off optimization problem to resolve conflicts between task and distillation objectives.

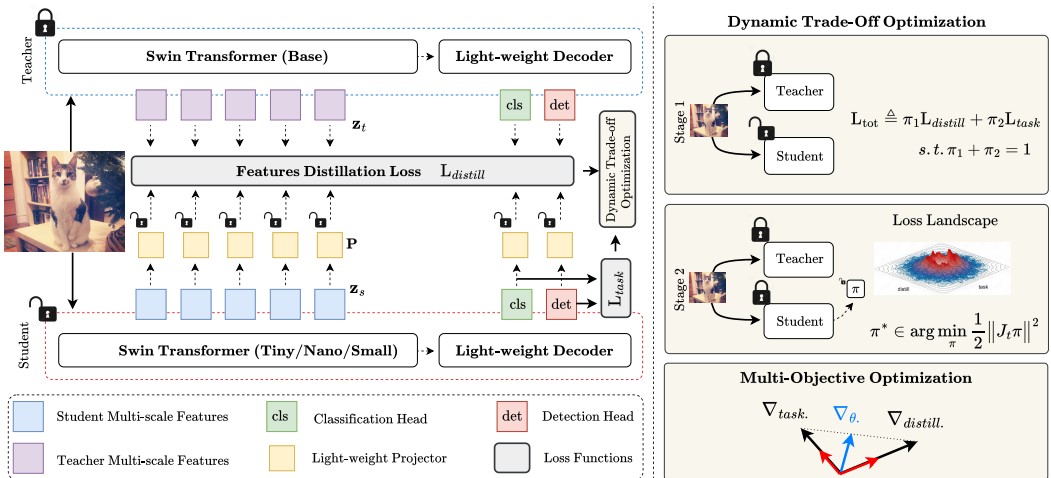

Figure 2: In DTO-KD, the teacher and student models simultaneously process the input image $\boldsymbol{x}$. Each network consists of a Swin Transformer with a lightweight decoder. The teacher's features ($\boldsymbol{z}_t$), and the student's ($\boldsymbol{z}_s$), are aligned using multiple lightweight projectors (P) at different scales. We formulate training as a multi-objective optimization (MOO) problem and propose a Dynamic Trade-off Optimization module that jointly minimizes the distillation loss $L_{\text{distill}}$ and the task-specific loss $L_{\text{task}}$, guiding them toward Pareto optimality.

## 3 METHOD

We introduce a Dynamic Trade-off Optimization for Knowledge Distillation (DTO-KD), with a specific focus on resolving the conflicting objectives in the KD process.

**Problem formulation.** We aim to transfer knowledge from a high-capacity teacher model with parameters $\boldsymbol{\phi}$, to a more compact model student, with parameters $\boldsymbol{\theta}$, focusing mainly on classification and detection tasks in visual recognition. We show the training data with $\mathcal{S} = \{(\boldsymbol{x}_i, \boldsymbol{y}_i)\}_{i=1}^{N}$, with $\boldsymbol{x}_i \in \mathbb{R}^d$ being the i-th input instance and $\boldsymbol{y}_i$ the corresponding target (*e.g.* a class label, bounding box). Our goal is to train the student model to effectively mimic the behavior of the teacher model over the dataset $\mathcal{S}$. Figure 2 shows an illustration of our proposed framework.

Effectively performing knowledge distillation requires balancing two objectives: the student must learn from two supervisory signals (e.g., one from the teacher and one from the task). We represent the teacher's loss as $L_{\text{distill}}$ and the task's loss as $L_{\text{task}}$. While we will define these more specifically for image classification and object detection in the appendix, we provide their general forms here:

$$L_{\text{distill}}(\boldsymbol{\theta}) \triangleq \mathbb{E}_{(\boldsymbol{x},\boldsymbol{y})\sim\mathcal{S}}\,\ell_{\text{distill}}\big(f_s(\boldsymbol{x};\boldsymbol{\theta}), f_t(\boldsymbol{x};\boldsymbol{\phi})\big) \tag{1}$$

$$L_{\text{task}}(\boldsymbol{\theta}) \triangleq \mathbb{E}_{(\boldsymbol{x},\boldsymbol{y})\sim\mathcal{S}}\,\ell_{\text{task}}\big(f_s(\boldsymbol{x};\boldsymbol{\theta}), f_t(\boldsymbol{x};\boldsymbol{\phi})\big) \tag{2}$$

The conventional KD approaches (*e.g.*, (Hu et al., 2024; Zheng & Yang, 2024)) train the student model by optimizing the loss as

$$L_{\text{tot}}(\boldsymbol{\theta}) \triangleq \alpha_1 L_{\text{distill}}(\boldsymbol{\theta}) + \alpha_2 L_{\text{task}}(\boldsymbol{\theta}) , \tag{3}$$

where $\alpha_1, \alpha_2 \in \mathbb{R}_+$ are the combination weights and hyperparameters of the model. The gradient of $L_{\text{tot}}(\boldsymbol{\theta})$ is

$$\boldsymbol{g}_{\text{tot}} = \nabla L_{\text{tot}}(\boldsymbol{\theta}) = \alpha_1 \boldsymbol{g}_{\text{dist}} + \alpha_2 \boldsymbol{g}_{\text{task}} \tag{4}$$

where $\boldsymbol{g}_{\text{dist}} = \nabla L_{\text{distill}}(\boldsymbol{\theta})$ and $\boldsymbol{g}_{\text{task}} = \nabla L_{\text{task}}(\boldsymbol{\theta})$ are the gradients of the distillation and task losses, respectively. Minimizing loss in Equation (3) for joint training introduces the following challenges:

**Gradient Conflict (GrC).** This occurs when the gradients of the distillation loss and the task loss conflict with each other. Mathematically, GrC happens when $\langle \boldsymbol{g}_{\text{dist}}, \boldsymbol{g}_{\text{task}} \rangle < 0$. During the optimization of the total loss $L_{\text{tot}}(\boldsymbol{\theta})$, the occurrence of GrC leads to conflicting gradient updates. Specifically, the total gradient $\boldsymbol{g}_{\text{tot}}$ may contradict either $\boldsymbol{g}_{\text{dist}}$ or $\boldsymbol{g}_{\text{task}}$, causing detrimental effects on one or both objectives. This conflict can exacerbate the learning dynamics, particularly in complex vision tasks such as object detection, by introducing unnecessary complexity into the training process.

**Gradient Dominance (GrD).** It arises when the gradients have significantly different magnitudes, leading one to dominate the update. When minimizing $L_{tot}(\boldsymbol{\theta})$, this imbalance may cause one objective to be completely neglected, as the update direction is primarily determined by the larger gradient, which can be estimated as $\frac{\|\boldsymbol{g}_{dist}\|}{\|\boldsymbol{g}_{task}\|}$. Lastly, tuning the hyperparameters $\alpha_1$ and $\alpha_2$ might become extremely tricky as the norm of gradients varies throughout optimization.

To address the aforementioned challenges, we advocate for the use of multi-objective optimization in KD. Specifically, we formulate the training process as optimizing the objective vector $L_{tot}(\boldsymbol{\theta}) = (L_{distill}(\boldsymbol{\theta}), L_{task}(\boldsymbol{\theta}))^\top$. The goal is to find a solution $\boldsymbol{\theta}^*$ on the Pareto front, *i.e.*, a solution that is not dominated by any other parameter vector $\tilde{\boldsymbol{\theta}}$. Formally, $\boldsymbol{\theta}^*$ is Pareto optimal if is no $\tilde{\boldsymbol{\theta}}$ such that

$$\begin{pmatrix} L_{distill}(\tilde{\boldsymbol{\theta}}) \\ L_{task}(\tilde{\boldsymbol{\theta}}) \end{pmatrix} \preceq \begin{pmatrix} L_{distill}(\boldsymbol{\theta}^*) \\ L_{task}(\boldsymbol{\theta}^*) \end{pmatrix} \tag{5}$$

The notation $\boldsymbol{a} \preceq \boldsymbol{b}$ here means that vector $\boldsymbol{a}$ achieves a lower value for all its elements simultaneously over $\boldsymbol{b}$. As we will discuss in the next section, formulating KD using the proposed algorithm addresses both GrC and GrD by aligning the gradients. Furthermore, the use of MOO mitigates the difficulty of hyperparameter tuning, as it eliminates the need to manually define $\alpha_1$ and $\alpha_2$.

### 3.1 KD AS A DYNAMIC TRADE-OFF OPTIMIZATION

Inspired by Liu et al. (2023), we followed a two stage approach for learning the optimal trade-off between conflicting objectives during the model training.

**Stage 1:** In stage 1 and at time $t$, we update the student model via $\boldsymbol{\theta}_{t+1} = \boldsymbol{\theta}_t - \eta \boldsymbol{g}_t$, where $\eta \in \mathbb{R}_+$ is the learning step size. We define the rate of improvement for the distillation and task losses as:

$$r_{dist}(\boldsymbol{g}_t) = \frac{L_{distill}(\boldsymbol{\theta}_t) - L_{distill}(\boldsymbol{\theta}_{t+1})}{L_{distill}(\boldsymbol{\theta}_t)},$$

$$r_{task}(\boldsymbol{g}_t) = \frac{L_{task}(\boldsymbol{\theta}_t) - L_{task}(\boldsymbol{\theta}_{t+1})}{L_{task}(\boldsymbol{\theta}_t)} . \tag{6}$$

In essence, $r_{dist}(\boldsymbol{g}_t)$ and $r_{dist}(\boldsymbol{g}_t)$ measure how much each loss can be improved by moving the parameters with $-\eta \boldsymbol{g}_t$. A larger value of $r_{dist}$ or $r_{task}$ implies the associated task has been improved more.

**Stage 2:** In stage 2, our goal is to determine an update $\boldsymbol{g}_t$ that maximizes the improvement over the worst-case rate. This can be achieved using a min-max optimization as:

$$\max_{\boldsymbol{g}_t \in \mathbb{R}^n} \min_{i \in \{dist, task\}} \frac{1}{\gamma} r_i(\boldsymbol{g}_t) - \frac{1}{2} \|\boldsymbol{g}_t\|^2 . \tag{7}$$

Here, $\gamma \in \mathbb{R}_+$ is a weighting hyperparameter. As shown in Liu et al. (2023), the solution of Equation (7) can be obtained via solving its dual problem as (see proposition 3.1 in Liu et al. (2023)). Define $\boldsymbol{\pi} = (\pi_1, \pi_2)^\top$ on the simplex $\Delta$ (*i.e.*, $\pi_1 + \pi_2 = 1, \pi_1, \pi_2 \geq 0$), and let $\boldsymbol{J}_t \in \mathbb{R}^{n \times 2}$ be

$$\boldsymbol{J}_t = \left[ \nabla \log \left( L_{distill}(\boldsymbol{\theta}_t) \right) \mid \nabla \log \left( L_{task}(\boldsymbol{\theta}_t) \right) \right]^\top \tag{8}$$

Then

$$\boldsymbol{\pi}_t^* \in \arg\min_{\boldsymbol{\pi} \in \Delta} \frac{1}{2} \|\boldsymbol{J}_t \boldsymbol{\pi}\|^2 , \tag{9}$$

and $\boldsymbol{g}_t = \boldsymbol{J}_t \boldsymbol{\pi}^* = \pi_1 \nabla \log \left( L_{distill}(\boldsymbol{\theta}_t) \right) + \pi_2 \nabla \log \left( L_{task}(\boldsymbol{\theta}_t) \right)$.

**Theoretical Properties.** The problem formulation in Equation (9) admits an analytical solution, unlike the general case studied in Liu et al. (2023). In this part, we establish key theoretical properties of the obtained update direction $\boldsymbol{g}^*$.

**Theorem 3.1** (Closed Form Solution). *Let* $\boldsymbol{J}_t = \left[ \nabla \log \left( L_{distill}(\boldsymbol{\theta}_t) \right), \nabla \log \left( L_{task}(\boldsymbol{\theta}_t) \right) \right] \in \mathbb{R}^{n \times 2}$. *The closed-form solution to the optimization problem*

$$\boldsymbol{\pi}^* \in \arg\min_{\boldsymbol{\pi}} \frac{1}{2} \left\| \boldsymbol{J}_t \boldsymbol{\pi} \right\|^2$$

$$s.t. \quad \pi_1 + \pi_2 = 1 \tag{10}$$

*is given by*

$$\pi_1^* = \frac{g_{22} - g_{12}}{g_{11} + g_{22} - 2g_{12}} \ , \tag{11}$$

$$\pi_2^* = \frac{g_{11} - g_{12}}{g_{11} + g_{22} - 2g_{12}} \ , \tag{12}$$

*where $\boldsymbol{G} = \boldsymbol{J}_t^\top \boldsymbol{J}_t$ is the Gram matrix:*

$$\boldsymbol{G} = \begin{bmatrix} g_{11} & g_{12} \\ g_{21} & g_{22} \end{bmatrix},$$

*with elements*

$$g_{11} = \left\| \nabla \log \left( L_{distill}(\boldsymbol{\theta}_t) \right) \right\|^2 , \tag{13}$$

$$g_{12} = g_{21} = \left\langle \nabla \log \left( L_{distill}(\boldsymbol{\theta}_t) \right), \nabla \log \left( L_{task}(\boldsymbol{\theta}_t) \right) \right\rangle , \tag{14}$$

$$g_{22} = \left\| \nabla \log \left( L_{task}(\boldsymbol{\theta}_t) \right) \right\|^2 . \tag{15}$$

The closed-form nature of this solution allows for efficient computation of the optimal weighting factors. One key property of the derived solution is that the update direction aligns with both objectives, ensuring that both the distillation and task losses are reduced simultaneously. This directly addresses GrC by preventing destructive interference between the two gradients.

**Corollary 3.2** (Alignment of $\boldsymbol{g}^*$). *Define $\boldsymbol{g}_1 = \nabla \log \left( L_{distill}(\boldsymbol{\theta}_t) \right)$ and $\boldsymbol{g}_2 = \nabla \log \left( L_{task}(\boldsymbol{\theta}_t) \right)$. Then the update direction $\boldsymbol{g}^* = \pi_1 \boldsymbol{g}_1 + \pi_2 \boldsymbol{g}_2$ for $\boldsymbol{\pi}^*$ defined in 11 is aligned with both $\boldsymbol{g}_1$ and $\boldsymbol{g}_2$.*

Another key property of the proposed solution is that it enforces equal contribution of the update direction to both gradients, effectively addressing GrD.

**Corollary 3.3** (Equal Contribution of $\boldsymbol{g}^*$ to Both Losses). *In Corollary 3.2, we showed that*

$$\langle \boldsymbol{g}^*, \boldsymbol{g}_1 \rangle = \langle \boldsymbol{g}^*, \boldsymbol{g}_2 \rangle = \frac{g_{11} g_{22} - g_{12}^2}{\| \boldsymbol{g}_1 - \boldsymbol{g}_2 \|^2}.$$

*This implies that the update direction contributes equally to the descent of both the distillation and task losses, effectively mitigating gradient dominance.*

An important aspect of any gradient-based optimization method is ensuring that update magnitudes remain within a controlled range to prevent vanishing or exploding gradients. Our solution satisfies both a lower and an upper bound on $\|\boldsymbol{g}^*\|$, ensuring stability during training.

**Corollary 3.4** (Lower Bound on $\|\boldsymbol{g}^*\|$). *The norm of the optimal update direction $\boldsymbol{g}^*$ satisfies the lower bound:*

$$\|\boldsymbol{g}^*\| \geq \frac{1}{\sqrt{2}} \min(\|\boldsymbol{g}_1\|, \|\boldsymbol{g}_2\|) \ . \tag{16}$$

*This implies that the update magnitude remains controlled and does not collapse under gradient imbalance.*

**Corollary 3.5** (Upper Bound on $\|\boldsymbol{g}^*\|$). *The norm of the optimal update direction $\boldsymbol{g}^*$ satisfies the upper bound:*

$$\|\boldsymbol{g}^*\| \leq \frac{\|\boldsymbol{g}_1\| \|\boldsymbol{g}_2\|}{|\|\boldsymbol{g}_1\| - \|\boldsymbol{g}_2\||}. \tag{17}$$

*As such, the magnitude of the updates does not grow excessively with different gradient scales.*

Finally, we observe that the algorithm's convergence is ensured by the general theoretical framework outlined in Liu et al. (2023). As our formulation aligns with it, the proposed optimization is guaranteed to converge to a Pareto optimal front.

**Practical Implementation.** The detailed algorithm for the proposed DTO-KD approach is detailed in Algorithm 1. The distillation and task weights $\boldsymbol{\pi}$ are initialized to 0.5. The algorithm begins by initializing the teacher as a frozen model and the student as a trainable model, and then extracts latent features from both for each training batch. The *DistillHead* and *TaskHead* refer to specific heads learning distillation and the task, respectively. It computes the distillation and task

---

**Algorithm 1** Dynamic Trade-off Optimisation for KD

---

1: **Inputs:** Dataset $\mathcal{S} = \{(\mathbf{x}_i, \mathbf{y}_i), ...\}$; Teacher $f_t$
2: **Initialise:** Student $f_s$ with $\theta$; Task weight $\pi_{distill} = \pi_{task} \leftarrow \frac{1}{2}$
3: **for** $t = 1 : T$ **do**                                                   *(iterations)*
4:      $\boldsymbol{x}_\tau, \boldsymbol{y}_\tau = \{(\boldsymbol{x}_b, \boldsymbol{y}_b)\}_{b=1}^B \sim S$                                     *(batch)*
5:      $\boldsymbol{z}_t, \boldsymbol{z}_s \leftarrow f_t(\boldsymbol{x}_\tau), f_s(\boldsymbol{x}_\tau)$                                 *(latent features)*
6:      $\hat{z}_t, \hat{z}_s \leftarrow \boldsymbol{z}_t^\top \boldsymbol{P} \boldsymbol{z}_s$                                       *(projection)*
7:      $\mathrm{L}(\boldsymbol{\theta}_t) = \begin{bmatrix} \mathrm{L}_{\text{distill}} \\ \mathrm{L}_{\text{task}} \end{bmatrix} = \begin{bmatrix} \ell_{distill}(DistillHead(\hat{z}_s, \hat{z}_t)) \\ \ell_{task}(TaskHead(z_s), y_\tau) \end{bmatrix}$            *(loss vector)*
8:      $\boldsymbol{g}_t = \pi_{distill} \nabla \log(\mathrm{L}_{\text{distill}}(\boldsymbol{\theta}_t)) + \pi_{task} \nabla \log(\mathrm{L}_{\text{task}}(\boldsymbol{\theta}_t))$
9:      $\boldsymbol{\theta}_{t+1} = \boldsymbol{\theta}_t - \gamma \boldsymbol{g}_t$                                  *(student model learning)*
10:     $\mathrm{L}(\boldsymbol{\theta}_{t+1}) \leftarrow f_s(\boldsymbol{x}_\tau)$                               *(frozen model inference)*
11:     $r(\boldsymbol{g}_t) = \begin{bmatrix} r_{\text{distill}}(\boldsymbol{g}_t) \\ r_{\text{task}}(\boldsymbol{g}_t) \end{bmatrix} = \begin{bmatrix} \frac{\mathrm{L}_{\text{distill}}(\boldsymbol{\theta}_t) - \mathrm{L}_{\text{distill}}(\boldsymbol{\theta}_{t+1})}{\mathrm{L}_{\text{distill}}(\boldsymbol{\theta}_t)} \\ \frac{\mathrm{L}_{\text{task}}(\boldsymbol{\theta}_t) - \mathrm{L}_{\text{task}}(\boldsymbol{\theta}_{t+1})}{\mathrm{L}_{\text{task}}(\boldsymbol{\theta}_t)} \end{bmatrix}$       *(update direction)*
12:     $\boldsymbol{\pi}(t+1) = \boldsymbol{\pi}(t) - \eta_\pi \nabla_{\boldsymbol{\pi}} \frac{1}{2} \left\| \pi_{distill}(t) \log(\mathrm{L}_{\text{distill}}(\boldsymbol{\theta}_t)) + \pi_{task}(t) \log(\mathrm{L}_{\text{task}}(\boldsymbol{\theta}_t)) \right\|^2$
                                                                               *(optimize task weights)*

---

losses, combines their gradients according to the current task weights, and updates the student model accordingly. After each update, the task weights are recalculated in closed form based on the relative improvement of each loss, ensuring a balanced optimization that aligns both the distillation and task objectives. Despite having strong theoretical properties, MTL algorithms (Liu et al., 2023), including the one we have developed above, require access to per task gradient, in our case access to $\boldsymbol{J} = [\nabla \log(\mathrm{L}_{\text{distill}}(\boldsymbol{\theta}_t)), \nabla \log(\mathrm{L}_{\text{task}}(\boldsymbol{\theta}_t))]$. This incurs performing two backpropagation per iteration, which is not desired. Instead, one can advocate to amortizing the training. This leads to an approximation to the algorithm while ensuring that an extra backprop step is not required. In short, the parameters $\boldsymbol{\pi} = (\pi_{distill}, \pi_{task})$ are updated via

$$\boldsymbol{\pi}(t+1) = \boldsymbol{\pi}(t) - \eta_\pi \nabla_{\boldsymbol{\pi}} \frac{1}{2} \left\| \pi_{distill}(t) \log(\mathrm{L}_{\text{distill}}(\boldsymbol{\theta}_t)) + \pi_{task}(t) \log(\mathrm{L}_{\text{task}}(\boldsymbol{\theta}_t)) \right\|^2 . \tag{18}$$

The update in Equation (18) does not guarantee $\boldsymbol{\pi} \in \Delta$, one should renormalize it via a softmax function. We have empirically observed that the amortized algorithm comfortably outperforms state-of-the-art KD algorithms with significant improvement over training speed. Specifically, the DTO-KD reaches the top performance of Roy Miles & Deng (2024) with 300 epochs in just 240 epochs.

## 4 EXPERIMENTS

We evaluate DTO-KD on two distinct vision tasks: image classification and object detection. For image classification, we adopt a CNN-based teacher model, RegNetY-160 (Radosavovic et al., 2020), and use transformer-based DeiT (Touvron et al., 2022) Small and Tiny as student models. For object detection, we employ transformer-based ViDT-Base (Song et al., 2021) as the teacher model, with ViDT-Small, ViDT-Tiny, and ViDT-Nano serving as the student models. Additionally, to assess the robustness of our method, we conduct distillation experiments using Vidt-Small as the teacher.

**Implementation details:** In DTO-KD, we reformulate model training as a gradient-based dynamic trade-off optimization problem. For the overall optimization across both classification and detection tasks, we use AdamW with a learning rate of 0.025 and a weight decay of 0.01. For classification, we adopt the training strategy and parameters from DeiT (Touvron et al., 2021a). Additionally, for data augmentation, we follow the method outlined in Roy Miles & Deng (2024). For learning, we employ AdamW (Loshchilov & Hutter, 2019) with a learning rate of 0.001 and a weight decay of 0.05. For object detection, we adhere to the training methodology from ViDT (Song et al., 2021). DTO-KD is trained using AdamW (Loshchilov & Hutter, 2019) with an initial learning rate of 10-4 for the body, neck, and head. We use the same hyperparameters as those in the ViDT (Song et al., 2021) transformer encoder and decoder. All experiments are conducted using PyTorch (Paszke et al., 2017) framework and executed on four NVIDIA H100 GPUs.

| Method | Venue | Top@1 | Teacher | #Param. |
|---|---|---|---|---|
| RegNetY-160 (Radosavovic et al., 2020) | *CVPR20* | 82.6 | None | 84M |
| CaiT-S24 (Touvron et al., 2021b) | *ICCV21* | 83.4 | None | 47M |
| DeiT3-B (Touvron et al., 2022) | *ECCV22* | 83.8 | None | 87M |
| DeiT-Ti (Touvron et al., 2021a) | *ICML21* | 72.2 | None | 5M |
| DeiT-Ti (KD) (Touvron et al., 2021a) | *ICML21* | 74.5 | Regnety-160 | 6M |
| ↳ 1000 epochs | *ICML21* | 76.6 | Regnety-160 | 6M |
| CivT-Ti (Ren et al., 2022) | *CVPR22* | 74.9 | Regnety-600m | 6M |
| Manifold (Hao et al., 2022) | *NeurIPS22* | 76.5 | CaiT-S24 | 6M |
| DearKD (Chen et al., 2022) | *CVPR22* | 74.8 | Regnety-160 | 6M |
| ↳ 1000 epochs | *CVPR22* | 77.0 | Regnety-160 | 6M |
| USKD (Yang et al., 2023) | *ICCV23* | 75.0 | Regnety-160 | 6M |
| MaskedKD (Son et al., 2024) | *ECCV24* | 75.4 | CaiT-S24 | 6M |
| SRD (Miles & Mikolajczyk, 2024) | *AAAI24* | 77.2 | Regnety-160 | 6M |
| $V_k$D-Ti (Roy Miles & Deng, 2024) | *CVPR24* | 78.3 | Regnety-160 | 6M |
| **DTO-KD (Ti)** | | **79.7** | Regnety-160 | 6M |
| DeiT-S (Touvron et al., 2021a) | *ICML21* | 79.8 | None | 22M |
| DeiT-S (KD) (Touvron et al., 2021a) | *ICML21* | 81.2 | Regnety-160 | 22M |
| ↳ 1000 epochs | *ICML21* | 82.6 | Regnety-160 | 22M |
| CivT-S (Ren et al., 2022) | *CVPR22* | 82.0 | Regnety-4gf | 22M |
| DearKD (Chen et al., 2022) | *CVPR22* | 81.5 | Regnety-160 | 22M |
| ↳ 1000 epochs | *CVPR22* | 82.8 | Regnety-160 | 22M |
| USKD (Yang et al., 2023) | *ICCV23* | 80.8 | Regnety-160 | 22M |
| MaskedKD (Son et al., 2024) | *ECCV24* | 81.4 | Deit3-B | 22M |
| SRD (Miles & Mikolajczyk, 2024) | *AAAI24* | 82.1 | Regnety-160 | 22M |
| $V_k$D-S (Roy Miles & Deng, 2024) | *CVPR24* | 82.3 | Regnety-160 | 22M |
| **DTO-KD (S)** | | **83.1** | Regnety-160 | 22M |

Table 1: **Object Classification task**: DTO-KD on the ImageNet-1K dataset. Unless specified, each model is only trained for 300 epochs.

## 4.1 OBJECT CLASSIFICATION USING IMAGENET-1K DATASET

We conducted extensive experiments on the ImageNet-1K dataset, using the RegNetY-160 (Radosavovic et al., 2020) model, pre-trained on the larger ImageNet-21K dataset, as the teacher to facilitate robust knowledge transfer. Two student models, DeiT-tiny and DeiT-small, were trained for 300 epochs on ImageNet-1K, and their performance was compared against existing state-of-the-art methods. As shown in Table 1, our approach demonstrates significant improvements in accuracy for both student models. Specifically, DTO-KD outperforms the baseline Touvron et al. (2021a) by 5.2 percentage points (pp) for the tiny model and 1.9 pp for the small model. Additionally, DeiT-tiny surpasses the previous state-of-the-art method Roy Miles & Deng (2024) by 1.4 pp, indicating a substantial enhancement in classification accuracy. For DeiT-small, our approach achieves a 0.8 pp accuracy improvement over Roy Miles & Deng (2024).

Additionally, compared to the baseline (Touvron et al., 2021a), which was trained for 1000 epochs, DTO-KD achieves a 3.1 percentage point (pp) improvement for the tiny model and a 0.5 pp improvement for the small model with just 300 epochs. This highlights the efficiency of our approach, demonstrating its ability to deliver competitive performance in significantly less training time, making it both effective and scalable.

## 4.2 OBJECT CLASSIFICATION USING CIFAR-100 DATASET

Conventional knowledge distillation methods are typically evaluated on both homogeneous and heterogeneous CNN architectures using the CIFAR-100 dataset. To position DTO-KD against these approaches, we benchmarked it following the protocols of prior KD works (Chen et al, 2021; Wang et al, 2024). Table 2 shows that DTO-KD also achieves superior results and reinforces its superiority, establishing a new SOTA by outperforming previous works in both small- and large-dataset settings.

## 4.3 OBJECT DETECTION

Table 3 demonstrates that our proposed method, DTO-KD, achieves state-of-the-art object detection performance on the MS-COCO benchmark (Lin et al., 2014), leveraging the ViDT transformer architecture (Song et al., 2022) for its strong performance and efficiency on consumer hardware. DTO-KD consistently improves upon various ViDT variants, enhancing the Swin-nano backbone by 0.7 percentage points (pp), Swin-tiny by 0.5pp, and Swin-small by 1.1pp. Notably, DTO-KD-

|  | **Homogeneous** | | | **Heterogeneous** | | |
| Methods | ResNet-56 ResNet-20 | WRN-40-2 WRN-40-1 | ResNet-32×4 ResNet-8×4 | ResNet-50 MobileNet-V2 | ResNet-32×4 ShuffleNet-V1 | ResNet-32×4 ShuffleNet-V2 |
|---|---|---|---|---|---|---|
| Teacher | 72.34 | 75.61 | 79.42 | 79.34 | 79.42 | 79.42 |
| Student | 69.06 | 71.98 | 72.50 | 64.60 | 70.50 | 71.82 |
| FitNet (Romero et al., 2015) | 69.21 | 72.24 | 73.50 | 63.16 | 73.59 | 73.54 |
| RKD (Park et al., 2019) | 69.61 | 72.22 | 71.90 | 64.43 | 72.28 | 73.21 |
| PKT (Passalis et al., 2020) | 70.34 | 73.45 | 73.64 | 66.52 | 74.10 | 74.69 |
| KD (Hinton et al., 2015) | 70.66 | 73.54 | 73.33 | 67.65 | 74.07 | 74.45 |
| OFD (Heo et al., 2019b) | 70.98 | 74.33 | 74.95 | 69.04 | 75.98 | 76.82 |
| CRD (Tian et al., 2019) | 71.16 | 74.14 | 75.51 | 69.11 | 75.11 | 75.65 |
| DIST (Huang et al., 2022) | 71.78 | 74.42 | 75.79 | 69.17 | 75.23 | 76.08 |
| ReviewKD (Chen et al., 2021) | 71.89 | 75.09 | 75.63 | 69.89 | 77.45 | 77.78 |
| DKD (Zhao et al., 2022b) | 71.97 | 74.81 | 75.44 | 70.35 | 76.45 | 77.07 |
| ReviewKD++ (Wang et al., 2024) | 72.05 | 75.66 | 76.07 | 70.45 | 77.68 | 77.93 |
| **DTO-KD (ours)** | **72.35** | **75.68** | **76.40** | **70.90** | **77.95** | **78.22** |

Table 2: **Object Classification task**: DTO-KD evaluated on both homogeneous and heterogeneous CNN architectures using the CIFAR-100 dataset.

|  | ViDT Model | Epochs | AP | $AP_{50}$ | $AP_{75}$ | $AP_S$ | $AP_M$ | $AP_L$ | #Params | FPS |
|---|---|---|---|---|---|---|---|---|---|---|
| **Teacher** | Swin-base (Song et al., 2021) | 50 | 49.4 | 69.6 | 53.4 | 31.6 | 52.4 | 66.8 | 0.1B | 9.0 |
| **Student** | Swin-nano (Song et al., 2021) | 50 | 40.4 | 59.6 | 43.3 | 23.2 | 42.5 | 55.8 | | |
|  | Token-Matching (Song et al., 2022) | 50 | 41.9 | 61.2 | 44.7 | 23.6 | 44.1 | 58.7 | 16M | 20.0 |
|  | $V_kD$-nano (Roy Miles & Deng, 2024) | 50 | 43.0 | 62.3 | 46.2 | 24.8 | 45.3 | 60.1 | | |
|  | **DTO-KD (nano)** | 50 | **43.7** | **63.1** | **46.8** | **25.1** | **46.2** | **61.9** | | |
|  | Swin-tiny (Song et al., 2021) | 50 | 44.8 | 64.5 | 48.7 | 25.9 | 47.6 | 62.1 | | |
|  | Token-Matching (Song et al., 2022) | 50 | 46.6 | 66.3 | 50.4 | 28.0 | 49.5 | 64.3 | 38M | 17.2 |
|  | $V_kD$-tiny (Roy Miles & Deng, 2024) | 50 | 46.9 | 66.6 | 50.9 | 27.8 | 49.8 | 64.6 | | |
|  | **DTO-KD (tiny)** | 50 | **47.4** | **67.2** | **51.3** | **28.0** | **50.7** | **65.8** | | |
|  | Swin-small (Song et al., 2021) | 50 | 47.5 | 67.7 | 51.4 | 29.2 | 50.7 | 64.8 | | |
|  | Token-Matching (Song et al., 2022) | 50 | 49.2 | 69.2 | 53.6 | 30.7 | 52.3 | 66.8 | 61M | 12.1 |
|  | $V_kD$-small (Roy Miles & Deng, 2024) | 50 | 48.5 | 68.4 | 52.4 | 30.8 | 52.2 | 66.0 | | |
|  | **DTO-KD (small)** | 50 | **49.6** | **69.4** | **53.9** | **31.6** | **53.1** | **67.1** | | |

Table 3: **Object Detection task**: Comparison with other detectors on COCO, with student models distilled from a pre-trained ViDT-base. Note that DTO-KD consistently outpeforms all challenging knowledge distillation baseline approaches.

small, with just 61M parameters, outperforms Swin-base (0.1B parameters) when both are trained from scratch. Additionally, DTO-KD-tiny, with 38M parameters, achieves nearly the same performance as Swin-small (61M parameters).

## 4.4 ABLATION STUDIES

**Impact of different components in DTO-KD:** We conduct a thorough evaluation of the impact of each primary component in DTO-KD, specifically assessing both stages of dynamic trade-off optimization, and post processing using gradient clipping. These components were introduced to enhance the knowledge distillation process, and their individual contributions are analyzed in Table 4. The results demonstrate that each stage significantly contributes to the performance of DTO-KD, with all showing a positive effect on the overall effectiveness of the model. Dynamic Trade-off opti-

| **Dynamic Trade-off Optimization** | | | **S:DTO-KD-nano / T:ViDT-base** | | |
| Proj | Optimization | Grad. Clip | AP | $AP_{50}$ | $AP_{75}$ |
|---|---|---|---|---|---|
| | | | 41.0 | 59.2 | 42.8 |
| | | ✓ | 41.8 | 61.2 | 44.7 |
| ✓ | | | 43.1 | 61.7 | 46.4 |
| ✓ | ✓ | | 43.6 | 62.9 | 46.6 |
| ✓ | ✓ | ✓ | **43.7** | **63.1** | **46.8** |

| **Student** | **ViDT-nano** | | **ViDT-tiny** | |
| **Teacher** | ViDT (small) | ViDT (base) | ViDT (small) | ViDT (base) |
|---|---|---|---|---|
| No Distillation (Song et al., 2021) | 40.4 | | 44.8 | |
| Token Matching (Song et al., 2022) | 41.5 | 41.9 | 45.8 | 46.5 |
| $V_kD$ (Roy Miles & Deng, 2024) | 42.2 | 43.0 | 45.9 | 46.9 |
| **DTO-KD (ours)** | **43.2** | **43.7** | **46.9** | **47.4** |

Table 4: **Component's Impact Assessment**: An ablation study showing the impact of projector and optimisation. We also applied gradient clipping as a pre-processing step to both objectives to see its impact with and without DTO.

Table 5: **Distillation from different teachers for the Object Detection task**: Comparison of ViDT on COCO2017 val set. We report AP for the student models distilled from different teacher models.

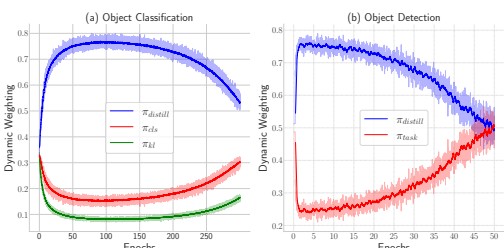 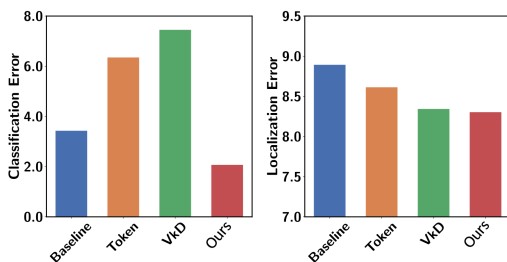

Figure 3: **Effectiveness of the Dynamic Balancing Strategy** on the object detection and the classification tasks.

Figure 4: **Error analysis:** Our dynamic trade-off optimisation approach consistently lowers the classification and localisation errors.

mization enables the model to handle diverse objectives, and alignment between teacher and student models to facilitate smoother knowledge transfer.

**Dynamic Balancing Strategy and $\pi$ values:** To better illustrate our approach, the figure below shows the varying weighting ratios of the distillation loss ($\pi_{distill}$) and the task losses ($\pi_{task}$) during training. As illustrated in **(a)**, we evaluate DTO-KD on a classification task using three distinct loss terms, demonstrating its ability to dynamically balance these objectives through adaptive weighting. In **(b)**, we extend this analysis to object detection with two loss terms, where DTO-KD's gradient-based vector optimization initially prioritizes the distillation loss and progressively shifts focus toward the task-specific loss.

**Subtask error analysis:** We conduct a thorough analysis of both classification and localization errors (Bolya et al., 2020) in the object detection task. DTO-KD outperforms other methods, achieving fewer errors in both areas while maintaining a strong balance between them. Notably, other KD techniques (Roy Miles & Deng, 2024; Song et al., 2022) underperform in the classification subtask compared to the baseline (Song et al., 2021), highlighting the superior effectiveness of our approach. See Figure 4 for more details.

**Distillation from different teachers:** Table 5 demonstrates DTO-KD's strong performance, even with smaller teachers like ViDT-small. This highlights its robustness, adaptability, and efficiency in resource-constrained settings, making it a versatile and effective distillation method across different teacher model scales.

## 5 LIMITATIONS

Like other KD methods, data availability is a bottleneck. DTO-KD is designed for distillation with available data, and extending it to data-free settings, especially for distilling from large pre-trained models, remains an open challenge. Extending DTO-KD to a data-free regime through sample synthesis may be more difficult due to its min-max optimization, which requires data for the training.

## 6 CONCLUSION

DTO-KD introduces a principled and effective solution to longstanding challenges in knowledge distillation, particularly for transformer-based architectures. By dynamically balancing task-specific and distillation objectives at the gradient level, DTO-KD mitigates supervision conflicts and gradient imbalances that arise from architectural mismatches between teacher and student models. This multi-objective formulation enables more stable and efficient training, resulting in student models that not only match but often exceed the performance of their non-distilled counterparts. Extensive evaluations on image classification and object detection benchmarks demonstrate that DTO-KD consistently achieves state-of-the-art results, setting a new standard for gradient-aware distillation methods. These improvements come with minimal computational overhead, making DTO-KD practical for real-world deployment.

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
