# DTO-KD: Dynamic Trade-off Optimization for Effective Knowledge Distillation Supplementary Material

**Zeeshan Hayder**[1,2], **Ali Cheraghian**[1,2], **Lars Petersson**[1], **Mehrtash Harandi**[3], **Richard Hartley**[2]
[1]Data61 / CSIRO, Australia
[2]Australian National University (ANU), Australia
[3]Monash University, Australia

## 1 Appendix

In this section, we provide a more in-depth theoretical analysis of the proposed DTO-KD optimisation approach.

### Theoretical Analysis of the Closed Form Formulation

In this section, we provide a detailed formulation for the closed-form solution.

**Theorem 1.1** (Closed Form Solution). *Let* $\boldsymbol{J}_t = [\nabla \log\left(L_{distill}(\boldsymbol{\theta}_t)\right), \nabla \log\left(L_{task}(\boldsymbol{\theta}_t)\right)] \in \mathbb{R}^{n \times 2}$. *The closed-form solution to the optimization problem*

$$\boldsymbol{\pi}^* \in \arg\min_{\boldsymbol{\pi}} \frac{1}{2}\left\|\boldsymbol{J}_t\boldsymbol{\pi}\right\|^2$$
$$s.t. \quad \pi_1 + \pi_2 = 1 \tag{1}$$

*is given by*

$$\pi_1^* = \frac{g_{22} - g_{12}}{g_{11} + g_{22} - 2g_{12}} \, , \tag{2}$$

$$\pi_2^* = \frac{g_{11} - g_{12}}{g_{11} + g_{22} - 2g_{12}} \, , \tag{3}$$

*where* $\boldsymbol{G} = \boldsymbol{J}_t^\top \boldsymbol{J}_t$ *is the Gram matrix:*

$$\boldsymbol{G} = \begin{bmatrix} g_{11} & g_{12} \\ g_{21} & g_{22} \end{bmatrix},$$

*with elements*

$$g_{11} = \left\|\nabla \log\left(L_{distill}(\boldsymbol{\theta}_t)\right)\right\|^2,$$
$$g_{12} = g_{21} = \left\langle \nabla \log\left(L_{distill}(\boldsymbol{\theta}_t)\right), \nabla \log\left(L_{task}(\boldsymbol{\theta}_t)\right)\right\rangle,$$
$$g_{22} = \left\|\nabla \log\left(L_{task}(\boldsymbol{\theta}_t)\right)\right\|^2.$$

*Proof.* Using the constraint $\pi_2 = 1 - \pi_1$, we can expand the objective function as follows:

$$\frac{1}{2}\left\|\boldsymbol{J}_t\boldsymbol{\pi}\right\|^2 = \frac{1}{2}\left\|\pi_1 \nabla \log\left(L_{distill}(\boldsymbol{\theta}_t)\right) + \pi_2 \nabla \log\left(L_{task}(\boldsymbol{\theta}_t)\right)\right\|^2$$
$$= \frac{1}{2}\left(\pi_1^2 g_{11} + \pi_2^2 g_{22} + 2\pi_1\pi_2 g_{12}\right)$$
$$= \frac{1}{2}\left(\pi_1^2 g_{11} + (1 - \pi_1)^2 g_{22} + 2\pi_1(1 - \pi_1)g_{12}\right)$$

Taking the derivative with respect to $\pi_1$ and setting it to zero:

$$\frac{d}{d\pi_1} \frac{1}{2} \left( \pi_1^2 g_{11} + (1-\pi_1)^2 g_{22} + 2\pi_1(1-\pi_1)g_{12} \right) =$$

$$\pi_1 \left( g_{11} + g_{22} - 2g_{12} \right) - g_{22} + g_{12}$$

$$\implies \boxed{\pi_1^* = \frac{g_{22} - g_{12}}{g_{11} + g_{22} - 2g_{12}}} .$$

Using the fact that $\pi_1 + \pi_2 = 1$, we obtain

$$\pi_2^* = \frac{g_{11} - g_{12}}{g_{11} + g_{22} - 2g_{12}} .$$

$\square$

**Corollary 1.2** (Alignment of $g^*$). *Define $g_1 = \nabla \log \left( L_{distill}(\theta_t) \right)$ and $g_2 = \nabla \log \left( L_{task}(\theta_t) \right)$. Then the update direction $g^* = \pi_1 g_1 + \pi_2 g_2$ for $\pi^*$ defined in Equation (2) is aligned with both $g_1$ and $g_2$.*

*Proof.* Assume that $g_1$ and $g_2$ are in conflict (*i.e.*, $-1 < \langle g_1, g_2 \rangle < 0$). We study the behavior of $\langle g^*, g_1 \rangle 0$ and $\langle g^*, g_2 \rangle$ below.

$$\langle g^*, g_1 \rangle = \langle \pi_1 g_1 + \pi_2 g_2, g_1 \rangle = \pi_1 \|g_1\|^2 + \pi_2 \langle g_2, g_1 \rangle$$

$$= \frac{g_{22} - g_{12}}{g_{11} + g_{22} - 2g_{12}} g_{11} + \frac{g_{11} - g_{12}}{g_{11} + g_{22} - 2g_{12}} g_{12}$$

$$= \frac{g_{11}g_{22} - g_{12}^2}{\|g_1 - g_2\|^2}$$

Similarly, for $g_2$:

$$\langle g^*, g_2 \rangle = \langle \pi_1 g_1 + \pi_2 g_2, g_2 \rangle = \pi_1 \langle g_1, g_2 \rangle + \pi_2 \|g_2\|^2$$

$$= \frac{g_{22} - g_{12}}{g_{11} + g_{22} - 2g_{12}} g_{12} + \frac{g_{11} - g_{12}}{g_{11} + g_{22} - 2g_{12}} g_{22}$$

$$= \frac{g_{11}g_{22} - g_{12}^2}{\|g_1 - g_2\|^2}.$$

Note that $g_{11}g_{22} - g_{12}^2 = \det(J^\top J)$ and since $G = J^\top J$ is always positive semi-definite, then $\det(J^\top J) \geq 0$. This proves that $g^*$ is always positively aligned with both $g_1$ and $g_2$. $\square$

**Corollary 1.3** (Equal Contribution of $g^*$ to Both Losses). *In Corollary 1.2, we showed that*

$$\langle g^*, g_1 \rangle = \langle g^*, g_2 \rangle = \frac{g_{11}g_{22} - g_{12}^2}{\|g_1 - g_2\|^2}.$$

*This implies that the update direction contributes equally to the descent of both the distillation and task losses, effectively mitigating gradient dominance.*

**Corollary 1.4** (Lower Bound on $\|g^*\|$). *The norm of the optimal update direction $g^*$ satisfies the lower bound:*

$$\|g^*\| \geq \frac{1}{\sqrt{2}} \min(\|g_1\|, \|g_2\|) . \tag{4}$$

*This implies that the update magnitude remains controlled and does not collapse under gradient imbalance.*

*Proof.* We have:

$$\|g^*\|^2 = \frac{\|g_1\|^2 \|g_2\|^2}{\|g_1 - g_2\|^2} .$$

Note that $\|g_1 - g_2\|^2 \leq \|g_1\|^2 + \|g_2\|^2$ and hence

$$\|g^*\|^2 \geq \frac{\|g_1\|^2\|g_2\|^2}{\|g_1\|^2 + \|g_2\|^2} \ .$$

Without loss of generality, assume $\|g_1\| \leq \|g_2\|$. We have

$$\|g^*\|^2 \geq \frac{\|g_1\|^2\|g_2\|^2}{\|g_1\|^2 + \|g_2\|^2}$$

$$\geq \frac{\|g_1\|^2\|g_2\|^2}{2\max\left(\|g_1\|^2, \|g_2\|^2\right)} = \frac{1}{2}\min\left(\|g_1\|^2, \|g_2\|^2\right)$$

As such,

$$\|g^*\| \geq \frac{1}{\sqrt{2}}\min(\|g_1\|, \|g_2\|) \ .$$

Thus, the update direction always maintains a lower bound, ensuring that it does not collapse even when one gradient is small. $\square$

**Corollary 1.5** (Upper Bound on $\|g^*\|$). *The norm of the optimal update direction $g^*$ satisfies the upper bound:*

$$\|g^*\| \leq \frac{\|g_1\|\|g_2\|}{|\|g_1\| - \|g_2\||}. \tag{5}$$

*As such, the magnitude of the updates does not grow excessively when the gradients have different scales.*

*Proof.* Note that

$$\|g_1 - g_2\|^2 = \|g_1\|^2 + \|g_2\|^2$$

$$-2\underbrace{\langle g_1, g_2\rangle}_{\leq \|g_1\|\|g_2\|} \geq (\|g_1\| - \|g_2\|)^2.$$

Hence

$$\|g^*\|^2 = \frac{\|g_1\|^2\|g_2\|^2}{\|g_1 - g_2\|^2} \leq \frac{\|g_1\|^2\|g_2\|^2}{(\|g_1\| - \|g_2\|)^2}$$

$$\implies \boxed{\|g^*\| \leq \frac{\|g_1\|\|g_2\|}{|\|g_1\| - \|g_2\||}} \ .$$

$\square$