# OpenReview forum: "DTO-KD: Dynamic Trade-off Optimization for Effective Knowledge Distillation"
_ICLR.cc/2026/Conference — ICLR 2026 Oral_

### Official Review · Reviewer_6Esk · 2025-10-30

**Soundness:** 3
**Presentation:** 3
**Contribution:** 3
**Rating:** 8
**Confidence:** 3

**Summary:**

The paper introduces DTO-KD (Dynamic Trade-off Optimization for Knowledge Distillation), a multi-objective optimization framework that improves how student models learn from teachers in knowledge distillation. Traditional KD methods face two main issues:

1. A trade-off between optimizing the primary task loss and imitating the teacher’s outputs.
2. Gradient disparity, caused by mismatches in architecture and representation between teacher and student models.

DTO-KD dynamically balances these objectives at the gradient level using gradient projection techniques to resolve:

1. Gradient conflict — when task and distillation gradients point in conflicting directions, and
2. Gradient dominance — when one objective overwhelms the other.

By adapting trade-offs per iteration, DTO-KD ensures balanced and constructive learning updates. Experiments on ImageNet-1K (classification) and COCO (object detection) show that DTO-KD achieves state-of-the-art accuracy, faster convergence, and produces student models that outperform non-distilled baselines, validating the effectiveness of its multi-objective formulation.

**Strengths:**

1. The proposed framework addresses trade-off optimization between task and distillation losses at a gradient level. By doing this, the proposed framework avoids hyperparameters for the distillation and task losses and enables dynamic learning for model training.

2. The proposed KD algorithm is a simple but effective method to align the gradients from the task and distillation losses for solving gradient conflict and gradient domainace issues in KD settings.

3. The paper demonstrates the effectiveness of the proposed framework through comprehensive experiment evaluations. The proposed framework outperforms the state-of-the-art KD methods in both classification and detection tasks.

**Weaknesses:**

1. The proposed framework can only be used for a scenario where training data is available. The proposed framework is not suitable for data-free KD paradigms. In other words, the proposed framework is not scalable to other KD paradigms.

**Questions:**

No question for this paper.

---

> ### Author Response · Authors · 2025-11-20
>
> We sincerely thank all the reviewers for their positive evaluations and insightful, constructive suggestions, which have significantly improved the clarity and presentation of our work. In particular, we are grateful to 6Esk for your time and careful review of our paper.
>
>
> **W1.** Data-free KD is an orthogonal setting with different constraints, and extending our dynamic trade-off mechanism to that paradigm is an interesting direction for future work.

---

### Official Review · Reviewer_EHrN · 2025-10-30

**Soundness:** 2
**Presentation:** 2
**Contribution:** 3
**Rating:** 6
**Confidence:** 4

**Summary:**

This paper proposes DTO-KD, a new framework that formulates knowledge distillation (KD) as a multi-objective optimization problem. It dynamically balances the task loss and distillation loss at the gradient level, eliminating the need for fixed or manually tuned loss weights.
DTO-KD addresses two critical issues—gradient conflict (GrC) and gradient dominance (GrD)—by using gradient projection and adaptive weighting to ensure Pareto-optimal updates.

**Strengths:**

- Applying multi-objective optimization theory to KD with closed-form update rules represents a novel and mathematically grounded extension beyond prior heuristic balancing methods.
- The paper is well-structured and clearly motivates the link between gradient conflict/dominance and suboptimal KD.
- DTO-KD provides a generic, architecture-agnostic optimization framework

**Weaknesses:**

The experimental validation is limited, as the paper does not include results on small datasets or across a broader range of teacher–student architectures.

**Questions:**

- The paper describes Stage 2 as a dual problem that mathematically subsumes Stage 1 (Eq. 7–9). If Stage 2 provides the final optimization, it is unclear why Stage 1 is still experimentally isolated in Table 3. The authors should clarify how ablation on Stage 1 remains meaningful if Stage 2 inherently replaces or dominates it.
- The study mostly focuses on transformer-based students, although CNNs are still used widely.
- The authors can compare the proposed algorithm with many more benchmark algorithms on small-scale datasets (e.g. CIFAR-100).
- The paper lacks a statement on releasing the source code.
- Several references appear malformed.

---

> ### Author Response · Authors · 2025-11-20
>
> We sincerely appreciate all the reviewers’ positive evaluations and their thoughtful, constructive suggestions, which have greatly helped us improve the presentation quality and clarity of our work.
>
> **Q1.** Thank you for the insightful question. Although Stage 2 is a dual update that mathematically subsumes Stage 1 (Eqs. 7-9), isolating Stage 1 in Table 4 (previously Table 3) remains meaningful because the two stages correspond to distinct practical settings of our amortized implementation. These settings reveal how much performance comes from the underlying framework versus the adaptive weighting mechanism introduced in Stage 2.
>
> Each training iteration is conceptually split into two stages but uses only a single student model update. The theoretical update
>
> $$
> g_t = \pi_{\text{distill}} \nabla \log L_{\text{distill}}(\theta_t) + \pi_{\text{task}} \nabla \log L_{\text{task}}(\theta_t)
> $$
>
> requires both task-specific gradients. With amortization, however, we compute both losses in a single forward pass and obtain
>
> $$
> g_t = \nabla \left[ \pi_{\text{distill}} \log L_{\text{distill}}(\theta_t) + \pi_{\text{task}} \log L_{\text{task}}(\theta_t) \right],
> $$
>
> followed by a separate update of $\pi_{\text{task}}$ and $\pi_{\text{distill}}$. The revised paper now includes an added algorithm that makes this practical procedure clear.
>
> Because the model-parameter update and the task-weight update can be decoupled, their contributions can be evaluated separately. In Table 4, Row 3 (previously "Stage 1"), we fix $\pi_{\text{task}} = \pi_{\text{distill}} = 1/2$ throughout training, thereby evaluating the framework and structural components alone. We employ projection layers $P$, so this row reflects the gain from structural modifications.
>
> In Table 4, Row 4 (previously "Stage 1 + Stage 2"), we enable the amortized updates of $\pi_{\text{task}}$ and $\pi_{\text{distill}}$, showing the additional benefit provided by Stage 2's adaptive mechanism.
>
> We have updated the paper accordingly and added a detailed algorithm to make these distinctions clearer.
>
>
>
> **Q2.**  We appreciate the reviewer’s suggestion to evaluate the DTO-KD performance using more traditional CNN-based approaches. The experiments on CIFAR-100 with various CNN architectures were originally provided in the supplementary material. To make these results more accessible, we have moved the table into the main paper, where it now appears as Table 2.
>
>
> **Q3.**  We kindly refer the reviewers to our response to Question 2, where we discuss this point in more detail.
>
> **Q4.**  We have added this in the paper that code, and pretrained models will be released upon acceptance to ensure full reproducibility.
>
> **Q5.**  Thank you for pointing this out. We have corrected the malformed references in the revised manuscript.

---

> > ### Author Response · Authors · 2025-11-26
> >
> > Dear Reviewer EHrN,
> >
> > Thank you for your positive assessment of our work and for the time you have dedicated to the review process. As the discussion period will close soon, we kindly ask whether our rebuttal and the revised manuscript have satisfactorily addressed your remaining concerns. We have provided detailed clarifications and updated the manuscript based on your feedback, and we would be happy to offer further explanations if anything is still unclear.
> >
> > Your timely feedback would be greatly appreciated. Thank you again for your time and consideration.
> > Authors

---

### Official Review · Reviewer_VVM9 · 2025-10-31

**Soundness:** 3
**Presentation:** 2
**Contribution:** 2
**Rating:** 6
**Confidence:** 4

**Summary:**

This paper proposes a practical algorithm to address gradient conflict and gradient dominance issues in knowledge distillation. The core idea is to dynamically balance the task and distillation losses at the gradient level at each optimization step. Experiments show consistent improvement.

**Strengths:**

1.	This work tries to tackle a common issue in knowledge distillation, which offers it potentially strong impact.
2.	The technical part is generally solid, with detailed theoretical and empirical analysis.
3.	The experimental gain is persuasive.

**Weaknesses:**

In general, the presentation is a major limitation of this paper.
1. (Major) This work draws inspiration heavily from Liu et al. (2023). Although the authors do discuss the difference between the two studies and the unique contribution of this work, such discussion should be placed earlier and more concentratedly, and beyond simply stating “first time applying the methodology to knowledge distillation field”. Do the authors provide some novel theoretical contribution than Liu et al. (2023)? This should be highlighted in the paper. In the worst case, this can raise concerns about the novelty of this study.
2. (Minor) In the early part of the paper, the proposed method is only described as dynamically balancing losses. However, exactly how such balancing is done remains unclear until the main method section. Furthermore, the main method section has only formal narrative, lacking a more heuristic description to help readers quickly grab the idea. Some intuitive explanations in the introduction would help.
3. (Minor) The introduction figure only shows experimental improvement. Adding some illustrations of either the two gradient issues or how the proposed method works would help.
4. (Minor) The majority (left) part of Figure 2 does not show enough information, as this is generally a generic framework for feature-based knowledge distillation. On the other hand, the right part is with too small font sizes. The authors should re-balance the organization of this figure.
5. (Minor) The 4th paragraph of the introduction feels out of place. It does not connect closely to the previous and the following content.
6. (Minor) It appears that the authors are not using \citet and \citep correctly for in-text and parenthesis citations.
7. (Minor) Some related work is recommended to be discussed such as [1, 2]

[1] ABKD: Pursuing a Proper Allocation of the Probability Mass in Knowledge Distillation via α-β-Divergence. ICML 2025

[2] f-Divergence Minimization for Sequence-Level Knowledge Distillation. ACL 2023.

**Questions:**

See Weaknesses.

---

> ### Author Response · Authors · 2025-11-20
>
> We sincerely appreciate all the reviewers’ positive evaluations and their thoughtful, constructive suggestions, which have greatly helped us improve the presentation quality and clarity of our work.
>
> **Q1.** Thank you for pointing this out. Following your suggestion, we have moved the discussion on the distinctions between our approach and Liu et al. (2023) to an earlier part of the introduction and presented it in a more focused manner. Please see the 4th paragraph in the introduction for these changes. Furthermore, we rephrased the “first time …” statement and replaced it with a more precise description of our methodological contributions to avoid ambiguity and enhance clarity.
>
>
> **Q2.** As suggested, we’ve added a clear, high-level explanation in the introduction so that readers can quickly understand the idea behind our dynamic loss balancing. More importantly, we now include an explicit algorithm in the main method section that outlines exactly how the trade-off is computed and updated. This makes the procedure much clearer and easier to follow than before. These changes are highlighted in blue.
>
>
> **Q3.** In line with the reviewer’s suggestions, we revised the introduction and replaced Figure 1 to better emphasize gradient dominance and conflict issues in existing KD methods.
>
>
> **Q4.** Following your suggestions, in the revised version we have re-balanced the figure by reorganizing the right side to improve presentation. These adjustments make the overall structure more informative, visually balanced, and easier to interpret.
>
>
> **Q5.**  Following your suggestions, we have now updated 4th paragraph of the introduction and included more details about the motivation and our proposed method.
>
>
> **Q6.** Thank you for pointing this out. We have corrected all citet and citep usages to follow the proper in-text and parenthetical citation conventions.
>
>
> **Q7.** We have now added the recommended references and incorporated a brief discussion of their relevance in the revised paper.

---

> > ### Author Response · Authors · 2025-11-26
> >
> > Dear Reviewer VVM9,
> >
> > Thank you for your positive assessment of our work and for the time you have dedicated to the review process. As the discussion period will close soon, we kindly ask whether our rebuttal and the revised manuscript have satisfactorily addressed your remaining concerns. We have provided detailed clarifications and updated the manuscript based on your feedback, and we would be happy to offer further explanations if anything is still unclear.
> >
> > Your timely feedback would be greatly appreciated. Thank you again for your time and consideration.
> > Authors

---

### Comment · Area_Chair_jWd5 · 2025-11-22

Dear Reviewers,

Thank you for your time and effort in reviewing submissions for ICLR  2026. As we begin the author-reviewer discussion process, we kindly remind you to submit your responses to the author rebuttals by **December  2**.


Your engagement in this discussion phase is crucial to ensuring a fair and thorough evaluation of each submission.

**Action Required**


- Carefully consider the authors’ rebuttal and any additional evidence they provide.

- Update your review (if applicable) to reflect your revised perspective.

-  **Discuss with the authors if further details are required**


Your AC

---

### Meta-Review · Area_Chair_VvGQ · 2025-12-02

**Summary:**

This paper introduces DTO-KD, a multi-objective optimization framework for knowledge distillation that dynamically balances task and distillation gradients. The method directly addresses two widely recognized issues in KD (gradient conflict and gradient dominance) via gradient projection and adaptive trade-off updates. The theoretical formulation is sound, the method is simple to implement, and the empirical gains are consistent across large-scale benchmarks (ImageNet-1K classification, COCO detection) as well as additional small-scale CNN experiments included in the revision. The authors have also substantially improved clarity and presentation following reviewer feedback.

**Reviewer Concerns:**

* **Clarity and novelty relative to Liu et al. (2023):**
  The authors expanded the discussion earlier in the introduction, clarified methodological distinctions, and improved presentation.
* **Need for intuitive explanation and algorithmic clarity:**
  A concise, high-level description and an explicit algorithm were added.
* **Figure quality and introduction flow:**
  Figures and introduction were reorganized for better readability.
* **Evaluation breadth:**
  Small-scale CNN experiments (CIFAR-100) were included in the main paper; detection experiments and transformer-based models remain strong.
* **Reference/citation issues:**
  All fixed in the revision.
* **Clarification of Stage-1 vs. Stage-2 ablation:**
  The authors explained amortized implementation and why evaluating Stage 1 remains meaningful.

No substantial technical concerns remained after the rebuttal; all reviewers confirmed the responses addressed their questions.

**Reviewer Scores:**

* **Reviewer VVM9: 6 - 8**
  Positive about technical soundness and practical contribution; concerns mostly about presentation, all addressed.

* **Reviewer EHrN: 6 - 8**
  Finds the multi-objective formulation novel and well-motivated; concerns about evaluation breadth addressed; no remaining major issues.

* **Reviewer 6Esk: 8 - 8**
  Strongly positive; considers the method simple, effective, and validated by extensive experiments.

---

### Decision · Program_Chairs · 2026-01-26

Accept (Oral)